# The Re-Addition of Seminal Plasma after Thawing Does Not Improve Buck Sperm Quality Parameters

**DOI:** 10.3390/ani11123452

**Published:** 2021-12-03

**Authors:** Uchechi Linda Ohaneje, Uchebuchi Ike Osuagwuh, Manuel Alvarez-Rodríguez, Iván Yánez-Ortiz, Abigail Tabarez, María Jesús Palomo

**Affiliations:** 1Department of Animal Medicine and Surgery, Universitat Autònoma de Barcelona, 08193 Bellaterra, Spain; uohaneje@gmail.com (U.L.O.); ucheosuagwuh@gmail.com (U.I.O.); ivan.yanez22@gmail.com (I.Y.-O.); 2Department of Theriogenology, Michael Okpara University of Agriculture, Umudike 0234, Nigeria; 3Department of Animal Health and Anatomy, Universitat Autònoma de Barcelona, 08193 Bellaterra, Spain; manuel.alvarez-rodriguez@liu.se; 4Department of Biomedical and Clinical Sciences, Linköping University, 58183 Linköping, Sweden; 5Faculty of Biological and Agricultural Sciences, University Veracruzana, Tuxpan 92850, Veracruz, Mexico; abi_tabarez@yahoo.com.mx

**Keywords:** cryopreservation, seminal plasma, incubation, buck sperm, season, age, melatonin

## Abstract

**Simple Summary:**

The *Cabra Blanca de Rasquera* is a native goat breed from Catalonia (Spain) facing the danger of extinction. To preserve its genetic diversity, ex situ conservation programs for semen cryopreservation and the development of a genetic sperm bank are of major importance. However, the post-thaw sperm quality in this species is still a major concern. Therefore, the aim of this research work was to establish an approach in the thawing protocol of cryopreserved sperm bank doses of this breed in order to achieve a better post-thaw sperm quality and improve reproductive outcomes through artificial insemination. Since seminal plasma has been found to provide a source of nutrients, improve sperm motility, and prevent premature activation during migration of spermatozoa in the female reproductive tract, our study proposed to assess the effect of the addition of 20% seminal plasma (SP) during a 3 h incubation period under in vitro conditions mimicking the in vivo survivability of thawed sperm, in relation to donor age, the season of collection, and melatonin implants of the males in the non-breeding season. However, the strategy of adding seminal plasma to the sperm after thawing failed to improve buck post-thawed sperm quality, thus further investigation is needed.

**Abstract:**

In order to achieve a higher post-thaw buck sperm quality, an approach in the thawing protocol of cryopreserved sperm doses under in vitro capacitation conditions mimicking the in vivo female environment was studied. Therefore, functional and kinetic characteristics of buck thawed sperm from males of different ages, the season of collection, and melatonin implanted males in the non-breeding season were assessed after 3 h of incubation in an in vitro fertilization (IVF) media with 20% of buck seminal plasma (SP). Previously, fresh ejaculates were collected via artificial vagina from eight males of the *Cabra Blanca de Rasquera* breed during two consecutive years in breeding and non-breeding periods. Prior to semen collection in non-breeding seasons, males were split into two groups: one group was implanted with melatonin, while the other was not. In each group, semen samples were pooled, centrifuged, and diluted in an extender containing 15% powdered egg yolk and 5% glycerol before freezing. After thawing, sperm were washed and incubated in three different media: (a) control media (modified phosphate-buffered saline (PBS), (b) IVF commercial media, and (c) IVF media + 20% SP. Sperm motility was evaluated by CASA, while plasma and acrosome membrane integrity, mitochondria activity, and DNA fragmentation were analysed by flow cytometer at 0 h and after 3 h incubation. A significant reduction in motility, mitochondrial activity, plasma, and acrosome membrane integrity were observed after incubation in the presence of SP, although similar to that observed in IVF media alone. DNA integrity was not affected under in vitro capacitation conditions, regardless of SP addition. In conclusion, the addition of SP failed to improve post-thaw buck sperm quality under in vitro conditions irrespective of male age, the season of collection, and melatonin implant.

## 1. Introduction

The use of frozen-thawed caprine semen for artificial insemination (AI) remains challenging due to its low fertility rate [1]. The fertilizing capacity of cryopreserved semen has been reported to be low due to the detrimental effect of freezing resulting in ultrastructural, functional, and biochemical damage as increased membrane permeability, hyper oxidation, and formation of reactive oxygen species and damages on the mitochondrial sheath and tail axoneme [2]. Traditionally, cryopreserving media contains widely used components such as egg yolk and glycerol, regarded as cryoprotectants, which help to minimise cryo-injuries during the freezing and thawing processes [3]. However, the seminal plasma of goat semen contains egg yolk coagulating enzymes (EYCE, phospholipase A2) from bulbourethral glands secretions that interact with egg yolk and produce toxic substances that affect sperm plasma membrane integrity and motility and reduce their freezing ability [4,5]. Therefore, in buck semen, prior to freezing, the removal of the seminal plasma by centrifugation should be considered necessary in order to minimise such deleterious effects caused by the cryopreservation procedure [6].

However, the importance of seminal plasma (SP) cannot be ignored due to its positive role in several key processes such as maintaining viability, providing resistance to cold shock, and preventing premature sperm activation within the female reproductive tract as well as improving the fertilizing ability of the spermatozoa [7,8]. Indeed, the addition of SP or its components into post-thawed sperm increases oxygen uptake and progressive motility, helps to recover some surface proteins, reverts cryodamage, and increases overall sperm quality parameters [8,9]. It has also been shown to be beneficial in acrosome and capacitation status in good semen freezer males [10] and in the chromatin, plasma, and acrosome membrane status in bad semen freezer males [11]. Conversely, the detrimental effects of SP on motility and viability of spermatozoa after freeze-thawing have also been reported [9].

On the other hand, for the maintenance of a good fertility ability of frozen semen after artificial insemination (AI) as well after in vitro fertilization (IVF), the longevity or intactness of sperm cell membrane and motion characteristics of thawed sperm is of utmost importance [12]. Osuagwuh and Palomo [13] reported that exposing frozen-thawed ram sperm to post-thaw incubation period may reveal some sublethal damages especially in the sperm plasma and acrosomal membranes which may not be visible immediately after thawing. Moreover, mammalian sperm undergo a series of biochemical and biophysical changes prior to fertilization, called capacitation, and this involves the removal of sperm coating material, cholesterol depletion, modification at the plasma membrane level, and changes in intracellular calcium among others [14]. Therefore, the incubation of thawed buck sperm under in vitro capacitation conditions mimicking the environment of the female reproductive tract after AI may broaden the knowledge of the effectiveness of the re-addition of SP on the behaviour of thawed sperm and in its potential fertility.

Lastly, male goats have been reported to be seasonal breeders, with better breeding performances seen during the autumn, considered as the breeding season [15]. Nevertheless, these adverse seasonality effects on semen production may be reversed or manipulated by using subcutaneous melatonin implants, especially during a non-breeding season by artificially mimicking day length effect [16]. Furthermore, semen quality can be influenced by donor age, indicating that sexual maturation and semen quality increase as a male gets older due to the increase in testicular size [17]. However, some reports have shown no difference in semen quality between yearlings and mature bucks [18].

Therefore, the final objective of the present study was to find a new method during the thawing process which can be applied in all semen bank doses regardless of the freezing protocol used, the donor age, or the season of semen collection in order to increase fertility rates after AI is performed. Traditionally, since egg yolk has been used to create a semen bank from different goat breeds, and the removal of SP previous to sperm freezing has become widespread, our hypothesis lies on the fact that the re-addition of seminal plasma (SP) after thawing may improve buck sperm resilience, thereby increasing their survival and quality under in vitro capacitation conditions regardless of the origin or freezing protocol of buck semen.

## 2. Materials and Methods

This study was carried out over 2 consecutive years, using eight male goats from the *Blanca de Rasquera* breed. Experimental animals were housed in the *Institut de Recerca i Tecnologia Agroalimentàries* (IRTA), located at *Caldes de Montbui* in the *Vallès Oriental* shire, in the province of Barcelona, Spain, at 203 m of altitude, 41°37′55.77″ north latitude and 2°10′3.12″ eastern longitude, under an intensive management system. Animal handling was performed in accordance with Spanish Animal Protection Regulation, RD 53/2013, which conforms to European Union Regulation 2010/63.

The experimental design for this study consisted as follows: fresh ejaculates were collected from the 8 males at the age of 12 months old in autumn (breeding season, n = 6 days of collection). A similar semen collection was repeated from the same males in the following autumn, at the age of 24 months old (breeding season, n = 6 days of collection). In spring (non-breeding season), ejaculates were also collected from the same 8 males when males were 18 and 30 months old (n = 6 days of collection/age). To this regard, and prior to the semen collection in the spring, males were equally and randomly divided into two groups (4/group): one group was implanted with a slow releasing implant containing 18 mg of melatonin (Melovine^®^, CEVA-Animal Health, Amersham, UK) which was placed at the base of the ear for 60 days prior to semen collection, while the other group had no implants. Thus, 6 different experimental groups were generated in order to assess the sperm functionality and characteristics after the freezing/thawing procedure as well as the effect of the addition of 20% seminal plasma [19] during a 3 h incubation under in vitro capacitation conditions, in relation to donor age, the season of collection, and melatonin implants of the males in the non-breeding season.

### 2.1. Reagents and Media

All chemicals and reagents used for this experiment were purchased from Sigma Chemical Co. (St. Louis, MO, USA). The powdered egg yolk was obtained from NIVE (Nunspeet Holland Eiproducten, Nunspeet, The Netherlands). The fluorescence probes and analysis kits ((LIVE/DEAD^®^ sperm viability kit (L-7011; PI and SYBR-14) and Mitotracker deep red (M22426)) were all purchased from Invitrogen SA (Barcelona, Spain). All fluorochromes were stored at −20 °C in the dark. The washing media used for single layer centrifugation (SLC) was Bovipure^®^ (Nidacon, Mölndal, Sweden) while the control incubation media used was a modified phosphate buffer solution (PBS solution supplemented with 36 μg/mL pyruvate and 0.5 mg/mL BSA with an osmolarity of 280–300 mOsm/kg and a pH of 7.3–7.4) and an in-vitro fertilization commercial media (BO-IVF, IVF Biosciences, Falmouth, UK).

### 2.2. Preparation of Extenders

The basic extender used in this study was a Tris hydroxymethylaminoethane-citric acid glucose (TCG) solution, composed of Tris (0.3 M), citric acid anhydrous (94.7 mM), and D (+)-glucose (27.75 mM) [20]. The pH and osmolarity were adjusted to 7.25 ± 0.05 and 333.0 ± 2.8 mOsm/kg respectively. Then, glycerol (5% *v*/*v*, final concentration) and antibiotics (1000 IU/mL sodium penicillin, 1.0 mg/mL streptomycin sulphate) were subsequently added, and the pH and osmolarity were finally adjusted to 7.0 ± 7.17 and 1327 ± 234 mOsm/kg, respectively. Finally, powdered egg yolk (Nunspeet Holland Eiproductep) was diluted (1:1.25) with Milli Q water and stirred for 20 min, and then the reconstituted powdered egg yolk was added to a final concentration of 15% (*v*/*v*) as published elsewhere [21].

### 2.3. Semen Collection, Processing, and Freezing

Semen collection was routinely performed via an artificial vagina twice per day and 2 days a week from all the males used in our study. All fresh ejaculates were evaluated macroscopically for volume using a transparent graduated tube. Mass motility and linear progressive motility were determined subjectively using a phase contrast microscope in a warm stage at 37 °C. Then, all fresh ejaculates collected were immediately pooled to eliminate individual differences, washed in TCG solution by centrifuging twice (600× *g*, 10 min), and diluted in the extender (15% powdered egg yolk and 5% of glycerol, final concentration). Thereafter, sperm concentration was adjusted to 400 × 10^6^ sperm/mL and equilibrated for 4 h at 5 °C before freezing in liquid nitrogen vapours at a distance of 5 cm for 10 min and finally submerged into the liquid nitrogen [21].

### 2.4. Thawing and Sperm Washing

After cryopreservation, two straws from each semen collection day (n = 6 replicates/group) from the 6 different experimental groups: autumn (12 and 24 months), spring (18 and 30 months/with or without melatonin) were thawed by immersion in a water bath at 37 °C for 30 s. Straws were cleaned, and the content was poured into a dry 1.5 mL tube and kept in a water bath (37 °C). Thereafter, each thawed sperm sample was diluted (1:4) with PBS to a concentration of 100 × 10^6^ sperm/mL. In order to remove the cryoprotectant from the thawed samples, a single layer centrifugation (SLC) procedure was performed. The colloid was brought to room temperature before use and diluted according to the manufacturer’s instruction: an 80% Bovipure^®^ solution was made by diluting with Bovidilute^®^ (Nidacon, Mölndal, Sweden), mixed thoroughly, and kept in a warm water bath (37 °C). Thereafter, 1 mL of the colloid solution was pipetted into a centrifuge tube and an aliquot of the extended semen (1 mL of sperm diluted in PBS containing 100 × 10^6^ sperm/mL) was layered on top carefully in a slanting position. After centrifugation at 300× *g* for 25 min, the supernatant was removed using a pipette. The pellet was diluted with 1 mL of Boviwash^®^ (Nidacon, Mölndal, Sweden) and centrifuged again at 300× *g* for 5 min. The supernatant was discarded and the sperm pellet was transferred to a clean tube and diluted with a modified PBS. All semen samples were evaluated immediately after sperm washing (SLC) at 0 h, and subsequently after a 3 h post-thawing incubation period.

### 2.5. Incubation of Washed Sperm Samples

Immediately after SLC/washing, all sperm samples from the various treatments/groups were incubated for 3 h in three different media: (a) modified PBS or control media, (b) in vitro fertilization commercial media (BO-IVF media, IVF Biosciences, UK), and (c) BO-IVF media + seminal plasma (20%). All seminal plasma samples were collected during the breeding season (autumn) from the same males used in this study. Briefly, an aliquot of pooled fresh semen was centrifuged at 10,000× *g* for 10 min at 5 °C. The supernatant was collected and centrifuged again at 10,000× *g* for 10 min at 5 °C to remove the remaining sperm and cell debris [22]. Afterward, the clear supernatant (SP) was recovered in an aliquot contained in 1.5 mL tubes and freeze-dried in liquid nitrogen before being stored at −80 °C. Prior to use, the SP was thawed in a warm water bath at 37 °C. For each treatment/group, sperm samples were diluted in the three incubation media at a concentration of 40 × 10^6^ sperm/mL and placed in an incubator (5% CO_2_) at 38.5 °C for 3 h.

### 2.6. Plasma and Acrosome Membrane Integrity and Mitochondrial Function Analysis

Plasma and acrosome membrane integrity, as well as mitochondrial function, were evaluated by flow cytometry, using a quadruple-staining technique as described elsewhere by Tabarez et al. [21]. The flow cytometry analysis was performed using the BD FACSCanto flow cytometer (BD Biosciences, San Diego, CA, USA), and samples were analysed using BD FACSDiva software (BD Biosciences, San Diego, CA, USA). The following fluorescent probes were used for this study: LIVE/DEAD^®^ sperm viability kit (SYBR-14 and Propidium Iodide (PI); L-7011, Invitrogen S.A., Barcelona, Spain) for plasma membrane integrity, Mitotracker deep red (M22426, Invitrogen S.A., Barcelona, Spain) for the detection of mitochondrial activity, and finally PE-PNA (GTX01509, Antibody Bcn, S.L., Barcelona, Spain) for acrosome integrity.

The analysis was performed using a final concentration of 1 nM of SYBR-14 (diluted in dimethyl sulfoxide (DMSO)), 1.5 µM of PI, 2.5 µg/mL PE-PNA (1 mg/mL of stock solution in a buffer composed of 3.0 M ammonium sulphate, 50 mM sodium phosphate, and 0.05% sodium acid, pH 7.0 containing 1 mM (Ca^2+^) and (Mn^2+^) ions), and 1.5 nM of Mitotracker deep red (diluted in DMSO) with 1 mL of diluted semen in PBS to a final sperm concentration of 1 × 10^6^/mL. Sperm samples were thoroughly mixed and incubated at 37 °C for 10 min in the dark to enable proper staining. Prior to analysis, sperm samples were re-mixed and sperm suspensions were subsequently run through a flow cytometer. Fluorescent probes SYBR-14, PE-PNA, and PI were excited in the flow cytometer using a 488 nm blue solid-state laser while the Mitotracker deep red was excited using a 633 nm He/Ne excitation laser. Dead cells were positive for PI, thus producing a red fluorescent signal which was detected using the 679LP filter detector (detects photons emitted at more than 670 nm wavelength). SYBR-14 positive cells with green-fluorescent signals were detected as live cells using the detector with the filter 530/30BP (detects photons emitted in the range of 515–545 nm wavelength). Cells with acrosome damage were positively stained by the PE-PNA emitting orange-fluorescent signal which was detected using the 585/42BP filter detector (detects photons emitted in the range of 564–650 nm). Sperm mitochondria function was detected with the Mitotracker deep red using a 660/20BP filter detector.

The following sperm population was taken into consideration after the analysis: (1) total viable sperm (SYBR14+/PI-); (2) viable cells with intact acrosome and active mitochondria (SYBR14+/PI-/PE-PNA-/Mitotracker+); (3) viable cells with damaged acrosome and active mitochondria (SYBR14+/PI-/PE-PNA+/Mitotracker+); (4) viable cells with intact acrosome and inactive mitochondria (SYBR14+/PI-/PE-PNA-/Mitotracker-); (5) viable cells with damaged acrosome and inactive mitochondria (SYBR14+/PI-/PE-PNA+/Mitotracker-); and (6) total damaged acrosome sperm (PE-PNA+).

### 2.7. Analysis of DNA Integrity: Sperm Chromatin Structural Assay (SCSA)

A DNA integrity assessment was performed immediately after SLC/washing and after 3 h of incubation in all semen groups/treatments using the technique described by Evenson et al. [23]. Briefly, sperm samples were diluted to a final concentration of 1–2 × 10^6^ sperm/mL in a TNE (0.01 M Tris-HCl, 0.15 M NaCl, 1Mm EDTA, pH 7.4) buffer solution. Thereafter, the samples were kept in aliquots of 1.5 mL microcentrifuge tubes, freeze-dried in liquid nitrogen, and stored at −80 °C until analysis. Prior to analysis, each aliquot was thawed for 2 min at 37 °C and kept on ice before staining. Acid–induced denaturation of DNA in situ was achieved by adding 200 µL of thawed sperm sample to 400 µL of acid detergent (0.1% (*v*/*v*) Triton X-100, 0.15 M NaCl, 0.08 M HCl, pH 1.2). At exactly 30 s, 1.20 mL of Acridine Orange (AO) staining solution (0.037 M citric acid, 0.126 M Na_2_HPO_4_, 0.0011 M sodium EDTA, 0.15 M NaCl, pH 6.0, 4 °C), containing 6 µg/mL of electrophoretically purified AO (Polysciences, Inc., Warrington, PA, USA) was added. Stained sperm samples were immediately kept on ice for 2 min and subsequently analysed by flow cytometer using the BD FACS Canto flow cytometer (BD Biosciences, San Diego, CA, USA). The flow cytometer was adjusted using standard sperm samples. At least 5000 events were recorded for each sample. The DNA fragmentation index (DFI) was calculated as the ratio of denatured single-stranded DNA (red colour) to the total cells acquired (red + green colour). The High DNA stainability (HDS) index was also evaluated, defined as the population with an elevated green value, outside the main population recorded.

### 2.8. Analysis of Sperm Motility and Kinetic Parameters

Sperm motion parameters were evaluated immediately after SLC/washing at 0 h and after 3 h incubation in all semen groups/treatments using the computer-assisted sperm analysis (CASA) system ISAS^®^ (PROISER S.L., Valencia, Spain). Briefly, sperm was diluted (1:10) in PBS, and a 5 µL drop of sperm suspension was placed on a preheated slide on a warm stage and covered with a 24 mm × 24 mm coverslip. Sperm motion parameters were assessed at 38 °C, at × 200 using a phase contrast microscope (Olympus BH-2, Tokyo, Japan). For each sample, at least five fields per drop were analysed and a minimum of 200 sperm cells were evaluated. Total motility (TM, %), progressive motility (PM, %), curvilinear velocity (VCL, µm/s), linear velocity (VSL, µm/s), mean velocity (VAP, µm/s), linearity coefficient (LIN = (VSL/VCL) × 100, %), straightness coefficient (STR = (VSL/VAP) × 100, %), amplitude of lateral head displacement (ALH, µm), and beat-cross frequency (BCF, Hz) were all evaluated. The settings used for the sperm image analyses were as follows: the number of images (25/s), optical (Ph-), scale (20× Olympus), particle area (>3 or <70 microns^2^), slow sperm (10–45 microns/s), average sperm (45–75 microns/s), rapid sperm (>75 microns/s), and progressive (80% STR). All samples and reagents were maintained at 37 °C.

### 2.9. Statistical Analysis

Statistical analyses were performed using the SAS statistical package (Version 9.4, 2015, SAS Institute Inc., Cary, NC, USA). A non-hierarchical multivariate cluster analysis was performed using the k-means model based on Euclidean distances calculated from kinematic parameters to identify motile sperm subpopulations within sperm samples, using the FASTCLUS procedure and the number of subpopulations was established using the elbow method, previously described by Bravo et al. [24]. The frequency distribution of the motile sperm subpopulations was performed for each replicate using the FREQ procedure. Then, a correction was made for the proportion of each subpopulation based on the total motility (TM) obtained from each sperm sample (sP = (sP/100) × TM), since the subpopulations resulting from the analysis correspond only to motile spermatozoa. In this way, a new subpopulation was obtained which precisely grouped the static spermatozoa (sP4 = 100% − TM). An ANOVA test was applied to compare the parameters of each sperm subpopulation, using the GLM (general linear model) procedure and the differences between means were analysed with a Tukey test (*p* < 0.05).

A one-way non-parametric procedure was applied to perform the Kruskal-Wallis test in order to compare different factors under study (season, age, melatonin treatment, and incubation media), using the NPAR1WAY procedure. With regards to the significant difference, pairwise comparisons were made by adjusting the *p*-value (<0.05) with the Bonferroni method. Results were expressed as means ± standard error of the mean (SEM).

## 3. Results

A significant reduction (*p* < 0.05) in total plasma membrane integrity percentages, as well as in the proportions of viable sperm with intact acrosome and active mitochondria were observed after 3 h incubation in IVF media regardless of the presence of SP (Table 1). The mean percentage of the total viable sperm with damaged acrosome and active mitochondria was low, though a significant increase was seen after 3 h incubation in IVF media regardless of the presence of SP in the autumn samples (Table 1). The proportions of viable sperm with inactive mitochondria (either with intact or damaged acrosome) were very low (<0.1%), hence the data are not shown.

Furthermore, samples collected in autumn showed a tendency of having a higher plasma membrane integrity under capacitation conditions compared to those collected in the spring regardless of melatonin implant. However, the total acrosome damage of thawed sperm after washing (T0) was statistically higher (*p* < 0.05) in the period of autumn compared to those collected in spring regardless of melatonin implant and male age, while after 3 h incubation, a significant increase was observed when samples were incubated in IVF media in the presence of SP with no differences between male age, season, or melatonin treatments (Table 1).

With regards to sperm DNA integrity, no effect was found on the high DNA stainability (HDS) index after 3 h incubation regardless of the media in any of the experimental groups, except in sperm samples from older males collected in autumn which showed a significant difference between control media and IVF media + SP. Similarly, there were no significant differences in the HDS index between male age, except for samples collected during spring from older males which showed a significant reduction compared to samples collected in autumn but not in all the treatments. Furthermore, the DNA fragmentation index (DFI) was not significantly affected after washing irrespective of age, season, and melatonin treatment (Table 2).

However, an increase in the DFI was recorded after 3 h incubation in PBS (control media) in all the different samples, except for those collected in the spring from 30 month old males. A similar increase in DFI was observed after 3 h incubation in IVF, with or without SP, during spring regardless of melatonin implant, meanwhile, autumn samples incubated in IVF media regardless of the presence of SP did not differ from samples analysed immediately after washing.

The statistical analysis of CASA parameters showed that no significant differences were found in the total motility (TM) and progressive motility (PM) immediately after washing and after incubation in the control media between sperm samples from different male ages, seasons, or melatonin treatment. However, after the 3 h incubation period, TM and PM were significantly reduced in most of the experimental groups, especially when SP was added to the media (Table 3). Furthermore, samples collected in autumn showed a higher TM and PM when incubated in IVF media compared to those in the spring regardless of male age or melatonin treatment, though no significant differences were observed amongst all the experimental groups.

With respect to sperm subpopulations, a total of 54,751 sperms were analyzed to establish motile subpopulations, observing that 12,445 sperm tracks presented some type of movement. By means of cluster analysis, four sperm subpopulations were obtained with different kinetic patterns (*p* < 0.05). The subpopulation 1 (sP1) sperm had moderately high velocities, LIN, and ALH, although the BCF was the highest compared to the other two subpopulations, while subpopulation 2 (sP2) presented the lowest velocities, ALH, and BCF, but the highest LIN. Subpopulation 3 (sP3) sperm presented the highest velocities and ALH, moderately high BCF, and the lowest LIN compared to the others. Finally, in this study, static sperm constituted the fourth motile subpopulation (sP4) (Table 4).

Significant reductions were observed in the sP1, sP2, and sP3 proportions after 3 h incubation in most of the experimental groups (*p* < 0.05). No differences were found between ages or treatment in the sP1 and sP2 sperm percentages except after incubation in IVF media, where samples collected in autumn had greater values (*p* < 0.05) compared to those in spring. Moreover, the percentage of sP3 was significantly higher just after washing (0 h) in samples collected in autumn. The sP4 significantly increased after 3 h incubation, whereas no differences were found across ages and season regardless of melatonin treatments, except in IVF media, where samples collected in autumn showed lower percentages of static sperm (Table 5).

## 4. Discussion

The functional and kinetic characteristics of cryopreserved buck sperm during a post-thaw incubation period, thus mimicking the environment of the female reproductive tract were explored. Moreover, the evaluation of SP addition to post-thawed buck sperm and its resilience under in vitro capacitation conditions with respect to male age, melatonin treatment, and season of semen collection was assessed, and our results showed various interactions herein.

In the present study, the incubation period in the control media had no negative effect on sperm plasma membrane integrity. However, a decrease in this parameter after incubation in IVF media regardless of the presence of SP was found, probably due to structural and functional changes in buck sperm under capacitation conditions [25]. In other words, the IVF media may have induced an increase in live acrosome reacted sperm which eventually dies if fertilization does not take place [26].

However, when seminal plasma was added to IVF media, this hypothesis may be more difficult to interpret, since SP has been reported to slow down the capacitation and acrosome reaction [27]. Notably, the incidence of an acrosome reaction observed just after the 3 h incubation period in IVF media with or without SP was low and similar across all ages and seasons regardless of melatonin treatment. Therefore, we may suggest that the commercial IVF media used was not efficient enough to induce in vitro capacitation and subsequently, acrosome reaction or, alternatively, the evidence of capacitation-like changes were not detected if it may have occurred before or after the end of the 3 h of incubation.

The high percentage of total acrosome damage (dead and viable sperm) seen in sperm incubated in IVF media in the presence of SP cannot be explained due to the incidence of spontaneous acrosome reaction, as discussed earlier. In this regard, a potential explanation may be that buck seminal plasma could cause acrosome damage due to the presence of a hydrolysing enzyme, phospholipase-A2, found in buck seminal plasma [28,29]. Nevertheless, our results are in agreement with a report in ram [30] demonstrating that the addition of SP did not improve the sperm plasma membrane integrity and acrosome status, despite the fact that seminal plasma has been reported to reverse cold shock [31,32]. This discrepancy between researchers may be due to the heterogeneity of seminal plasma composition which has been demonstrated across species [33].

On the other hand, the survival of sperm samples collected in autumn were slightly higher after 3 h incubation under in vitro capacitation conditions compared to those collected in spring regardless of melatonin implants, which could be related to seasonal effects showing higher semen quality on bucks during autumn [34,35,36]. However, total acrosome damage was surprisingly lower (*p* < 0.05) in samples collected in spring just after washing and after 3 h incubation in control media. This finding could be related to physiological or structural differences of sperm membranes during the non-breeding season [37], although age and season had no identifiable effect on the acrosome integrity after incubation in IVF media.

DNA integrity has been shown to be a reliable indicator of sperm quality in most animal species. In our study, the age of males, the season of semen collection, or the melatonin treatment appeared to not affect the extent of the DNA fragmentation after washing. However, an increase was observed after 3 h incubation in control media and is in agreement with Crespo et al. [38]. However, thawed samples collected in autumn and incubated in IVF media where DNA integrity was maintained were suggested to be due to the better quality of the semen during the breeding season [34]. In this regard, the post-thaw incubation period may have revealed some information not visible immediately after thawing which could partially explain the reason why semen produced in autumn could give better fertility rates. Despite reports on the antioxidant properties of the SP preventing the degradation of sperm DNA [39], our results showed no beneficial effect of its presence in the incubation media on the sperm DNA status.

Although no huge differences in high DNA stainability were observed between male ages regardless of season or melatonin treatment, a slightly decreasing tendency was observed in samples collected in spring. This seasonal effect agrees with a similar report showing higher HDS values in the breeding season when compared to the non-breeding season [40]. However, the HDS sperm population has also been described as immature cells [23], expecting that these peculiarities may be more related to sperm in the non-breeding season. Nevertheless, further investigation is required since this high DNA stainability in the buck sperm may reflect an unknown aspect of sperm function. No significant effect of incubation was seen on the HDS index, although Peris et al. [41] demonstrated that incubation of ram sperm enhanced sperm damage, thus leading to high values of DFI and HDS reflected by DNA strand breaks or poor chromatin condensation.

With regards to motility, a significant reduction was seen after 3 hrs incubation in all the media used, especially in IVF media in presence of SP, regardless of age, season, and melatonin treatment which may be due to mitochondria aging during cryopreservation resulting in low ATP production [42]. While the intactness of the plasma membrane was not significantly affected by 3 h incubation in the control media, motility was reduced, showing that cryopreservation may have different effects on sperm functionality with respect to different post-thawing methods [5].

Nevertheless, our results showed that TM and PM were not affected by the incubation media as much as seen in the sperm plasma membrane integrity. This implies that the in vitro capacitation conditions may have affected the sperm membrane integrity much more than the ability of the sperm to be motile [43]. Furthermore, this deleterious effect on sperm motility was more evident in spring samples which showed lower values regardless of melatonin treatment. Again, the application of a resilience test to analyse certain aspects which may not have been detected immediately after thawing is strongly recommended.

Furthermore, no improvement in sperm motility was observed in the presence of SP regardless of age, season, or melatonin treatment. On the contrary, the addition of SP or its components to ram post-thawed semen has been reported to increase sperm motility [30]. As mentioned earlier, differences in the sperm physiology and SP composition between species, as well the complex composition and inconsistency of SP effect on sperm have been demonstrated [19,44,45]. Nevertheless, similar reductions were seen in sperm incubated in IVF media alone, which was used as capacitation media in this study. It is worthy to note that buck sperm cells may lose their motility after incubation in an in vitro capacitation media once they reveal a “jerk” movement [46].

In the subpopulation structure, four different subpopulations were described in our study, including the non-motile sperm subpopulation or statics, but no significant changes in their distribution amongst ages, season, or melatonin treatment were found, except after IVF incubation where sP1 and sP2 proportions were significantly lower in spring while the static proportion (sP4) increased. More evident was the effect of 3 h incubation causing the increase in sP4 (statics) and the reduction in the sP1 proportions. Moreover, a further decrease in sP1 was observed in sperm incubated under capacitation conditions in the presence of SP, which may also be attributed to the detrimental effect of SP as earlier discussed. A similar effect was seen in the sP2 and sP3 subpopulations after 3 h incubation compared to the proportions just after washing. It is also worthy to note that sP3 may require special attention as this subpopulation exhibited some sperm motility characteristics (very high VCL and ALH and very low LIN) which may depict “hyperactivation” as described by Mortimer and Mortimer [47]. Our results showed that this “hyperactivated” subpopulation was generally lower in spring and almost absent when seminal plasma was added to the media, which could be related to the ability of SP to slow down capacitation, acrosome reaction, and hyperactivation [27] since these three phenomena occurred concurrently [48].

## 5. Conclusions

Buck seminal plasma did not improve motility and plasma membrane integrity of thawed buck spermatozoa under in vitro capacitation conditions irrespective of male age, season, or melatonin treatment. The addition of SP in the incubation media increased acrosome damage while the incidence of acrosome reaction was similar when IVF media alone was used. Furthermore, DNA integrity was maintained in vitro capacitation media regardless of the presence of SP during the breeding season. Finally, the role of seminal plasma is still controversial due to its complex composition in bucks, and also it is important to carry out a thermo-resistance test to reveal damages that may not be detected immediately after thawing.

## Figures and Tables

**Table 1 animals-11-03452-t001:** Effect of seminal plasma and its resilience under in-vitro capacitation conditions on the viability of thawed buck sperm with respect to male age (months) season, and melatonin implant.

Sperm Parameter (%)	Season of Collection	Autumn		Spring		Spring + Melatonin	
	Male Age (Months)	12	24	18	30	18	30
	Incubation Media						
Total plasma membrane integrity	T_0_	38.4 ± 3.7 ^1^	39.7 ± 3.0 ^1^	33.0 ± 5.1 ^1^	29.8 ± 5.7 ^1^	34.2 ± 5.5 ^1^	30.8 ± 4.7 ^1^
	Control	33.4 ± 1.7 ^1^	31.6 ± 2.7 ^1^	34.5 ± 4.7 ^1^	21.5 ± 4.7 ^1^	30.6 ± 4.3 ^1^	25.6 ± 4.1 ^1^
	IVF	13.1 ± 2.4 ^ab,2^	15.3 ± 2.9 ^a,2^	10.1 ± 2.3 ^abc,2^	4.1 ± 0.9 ^c,2^	7.1 ± 2.1 ^abc,2^	5.5 ± 1.9 ^bc,2^
	IVF + SP	19.0 ± 4.7 ^a,2^	15.9 ± 1.8 ^a,2^	6.3 ± 2.0 ^ab,2^	2.7 ± 0.8 ^b,2^	4.4 ± 1.4 ^b,2^	2.9 ± 0.6 ^b,2^
Intact acrosome and active mitochondrial sperm	T_0_	37.0 ± 3.8 ^1^	39.4 ± 3.0 ^1^	32.6 ± 5.2 ^1^	29.4 ± 5.7 ^1^	33.9 ± 5.4 ^1^	30.7 ± 4.7 ^1^
	Control	32.0 ± 1.8 ^1^	29.9 ± 3.2 ^1^	33.7 ± 4.6 ^1^	21.0 ± 4.6 ^1^	30.2 ± 4.2 ^1^	25.1 ± 4.0 ^1^
	IVF	11.6 ± 2.1 ^ab,2^	13.5 ± 2.2 ^a,2^	9.9 ± 2.3 ^abc,2^	3.5 ± 0.7 ^c,2^	6.7 ± 2.1 ^abc,2^	5.1 ± 1.9 ^bc,2^
	IVF + SP	18.3 ± 4.8 ^a,2^	14.6 ± 2.1 ^a,2^	5.6 ± 2.2 ^ab,2^	2.4 ± 0.7 ^b,2^	4.0 ± 1.4 ^b,2^	2.7 ± 0.7 ^b,2^
Damaged acrosome and active mitochondrial sperm	T_0_	0.0 ± 0.0 ^b,2^	0.1 ± 0.0 ^ab,2^	0.0 ± 0.0 ^b^	0.2 ± 0.1 ^a,12^	0.0 ± 0.0 ^b,2^	0.0 ± 0.0 ^b,2^
	Control	0.0 ± 0.0 ^2^	0.0 ± 0.0 ^2^	0.0 ± 0.0	0.0 ± 0.0 ^2^	0.0 ± 0.0 ^2^	0.1 ± 0.1 ^12^
	IVF	0.5 ± 0.2 ^ab,1^	0.9 ± 0.4 ^a,1^	0.1 ± 0.0 ^b^	0.3 ± 0.1 ^ab,1^	0.2 ± 0.1 ^ab,12^	0.4 ± 0.2 ^ab,1^
	IVF + SP	0.4 ± 0.2 ^ab,1^	1.0 ± 0.5 ^a,1^	0.1 ± 0.0 ^b^	0.3 ± 0.2 ^ab,1^	0.4 ± 0.2 ^ab,1^	0.2 ± 0.1 ^ab,12^
Total Acrosome Damage	T_0_	25.0 ± 4.7 ^ab,12^	35.6±3.2 ^a,12^	6.0 ± 0.8 ^c,2^	6.5 ± 0.9 ^c,23^	14.2 ± 3.0 ^bc,12^	8.5 ± 2.2 ^c,2^
	Control	20.6 ± 4.2 ^a,2^	19.1 ± 2.2 ^a,2^	13.3 ± 6.5 ^ab,12^	5.0 ± 0.7 ^b,3^	8.8 ± 2.2 ^ab,2^	8.6 ± 2.5 ^ab,2^
	IVF	21.1 ± 4.5 ^2^	28.0 ± 6.9 ^2^	14.8 ± 5.4 ^12^	15.0 ± 4.9 ^12^	15.3 ± 3.0 ^12^	19.5 ± 7.4 ^12^
	IVF + SP	58.0 ± 8.6 ^1^	52.9 ± 5.0 ^1^	49.3 ± 13.3 ^1^	50.7 ±12.3 ^1^	52.9 ± 12.8 ^1^	50.4 ±10.7 ^1^

^a–c^ Different letters represent significant differences (*p* < 0.05) between treatments and age. ^1–3^ Different numbers represent significant differences (*p* < 0.05) between samples within the parameters. T_0_ = time 0, just after sperm washing by single layer centrifugation (SLC); control = modified PBS, IVF: 3 h incubation in in vitro fertilization media; IVF+SP: 3 h incubation in in vitro fertilization media + 20% seminal plasma. Data are shown as mean ± S.E.M.

**Table 2 animals-11-03452-t002:** Effect of seminal plasma and its resilience under in vitro capacitation conditions on the DNA integrity of thawed buck sperm with respect to male age (months) season, and melatonin implant.

Sperm Parameter (%)	Season	Autumn		Spring		Spring + Melatonin	
	Male Age (Months)	12	24	18	30	18	30
	Incubation Media						
HDS	T_0_	4.7 ± 0.9 ^a^	4.2 ± 0.7 ^a,12^	2.4 ± 0.6 ^ab^	1.1 ± 0.5 ^b^	1.8 ± 0.4 ^ab^	2.0 ± 0.8 ^ab^
	Control	4.8 ± 0.9 ^a^	5.2 ± 1.2 ^a,1^	2.7 ± 0.5 ^ab^	2.0 ± 0.8 ^ab^	1.7 ± 0.4 ^ab^	1.2 ± 0.3 ^b^
	IVF	4.4 ± 1.2 ^a^	3.7 ± 0.8 ^a,12^	1.4 ± 0.4 ^ab^	1.6 ± 0.6 ^ab^	1.5 ± 0.5 ^ab^	0.8 ± 0.2 ^b^
	IVF + SP	3.0 ± 0.8 ^a^	1.6 ± 0.2 ^ab,2^	1.9 ± 0.5 ^ab^	1.1 ± 0.6 ^b^	1.5 ± 0.3 ^ab^	0.4 ± 0.1 ^b^
DFI	T_0_	5.9 ± 0.6 ^2^	6.8 ± 1.1 ^2^	7.1 ± 1.8 ^2^	12.9 ± 5.2	5.8 ± 0.5 ^2^	5.9 ± 1.1 ^2^
	Control	31.6 ± 3.2 ^a,1^	29.1 ± 5.0 ^ab,1^	21.2 ± 5.1 ^ab,1^	13.9 ± 1.7 ^b^	22.8 ±3.7 ^ab,1^	19.7 ±3.0 ^ab,1^
	IVF	6.5 ± 1.2 ^b,2^	6.1 ± 1.4 ^b,2^	17.7 ± 3.3 ^a,12^	15.9 ± 2.5 ^a^	18.9 ± 3.0 ^a,1^	17.9 ± 2.5 ^a,1^
	IVF + SP	7.5 ± 1.1 ^b,2^	7.2 ± 0.9 ^b,2^	15.0 ± 3.2 ^ab,12^	18.4 ± 2.5 ^ab^	20.7 ± 3.7 ^a,1^	18.5 ± 3.3 ^ab^

^a,b^ Different letters represent significant differences (*p* < 0.05) between treatments and age. ^1,2^ Different numbers represent significant differences (*p* < 0.05) between samples within the parameters. T_0_ = time 0, just after sperm washing by single layer centrifugation (SLC); control = modified PBS, IVF: 3 h incubation in vitro fertilization media; IVF+SP: 3 h incubation in vitro fertilization media + 20% seminal plasma, HDS: high DNA stainability. DFI: DNA fragmentation index. Data are shown as mean ± S.E.M.

**Table 3 animals-11-03452-t003:** Effect of seminal plasma and its resilience under in vitro capacitation conditions on the total and progressive motility of thawed buck sperm with respect to male age (months) season, and melatonin implant.

Sperm Parameter (%)	Season	Autumn		Spring		Spring + Melatonin	
	Male Age (Months)	12	24	18	30	18	30
	Incubation Media						
TM	T_0_	42.0 ± 4.1 ^1^	50.5 ± 2.9 ^1^	34.0 ± 5.1 ^1^	33.2 ± 3.9 ^1^	36.2 ± 3.7 ^1^	39.8 ± 4.8 ^1^
	Control	22.4 ± 3.1 ^2^	21.6 ± 3.3 ^23^	18.8 ± 2.1 ^12^	18.0 ± 2.1 ^2^	13.8 ± 2.4 ^2^	20.4 ± 5.4 ^12^
	IVF	24.1 ± 0.8 ^ab,2^	32.7 ± 4.2 ^a,12^	17.3 ± 3.5 ^abc,12^	15.4 ± 1.1 ^bc,2^	14.4 ± 2.8 ^bc,2^	10.0 ± 2.9 ^c,2^
	IVF + SP	18.5 ± 8.0 ^2^	16.1 ± 1.7 ^3^	15.3 ± 3.3 ^2^	11.0 ± 0.9 ^3^	11.4 ± 3.5 ^2^	8.9 ± 3.2 ^2^
PM	T_0_	17.2 ± 2.6 ^1^	16.0 ± 2.6 ^1^	10.5 ± 1.6	15.5 ± 1.2 ^1^	12.9 ± 1.4 ^1^	17.6 ± 2.4 ^1^
	Control	13.9 ± 3.0 ^12^	9.5 ± 1.5 ^12^	6.4 ± 1.1	8.6 ± 1.0 ^2^	6.2 ± 1.4 ^2^	8.5 ± 1.7 ^12^
	IVF	13.1 ± 0.9 ^a,12^	18.1 ± 3.4 ^a,1^	6.6 ± 1.5 ^b^	6.3 ± 1.3 ^b,23^	5.6 ± 1.2 ^b,2^	4.0 ± 1.6 ^b,2^
	IVF + SP	5.9 ± 1.9 ^ab,2^	8.3 ± 0.5 ^a,2^	4.6 ± 0.9 ^ab^	4.1 ± 0.5 ^ab,3^	4.2 ± 1.1 ^ab,2^	3.6 ± 1.2 ^b,2^

^a–c^ Different letters represent significant differences (*p* < 0.05) between treatments and age. ^1–3^ Different numbers represent significant differences (*p* < 0.05) between samples within the parameters, T_0_ = time 0, just after sperm washing by single layer centrifugation (SLC); control = modified PBS, IVF: 3 h incubation in in vitro fertilization media; IVF+SP: 3 h incubation in in vitro fertilization media + 20% seminal plasma. TM: total motility, PM: progressive motility. Data are shown as mean ± S.E.M.

**Table 4 animals-11-03452-t004:** Kinematic characteristics of the four sperm subpopulations in thawed buck sperm samples.

Kinematic Parameter	sP1	sP2	sP3	sP4
VCL (μm/s)	109.5 ± 21.9 ^b^	46.9 ± 23.4 ^c^	163.2 ± 23.5 ^a^	N.D.
VSL (μm/s)	41.9 ± 32.1 ^b^	20.6 ± 20.0 ^c^	46.5 ± 32.8 ^a^	N.D.
VAP (μm/s)	65.9 ± 30.0 ^b^	30.3 ± 20.8 ^c^	88.1 ± 33.0 ^a^	N.D.
LIN (%)	36.4 ± 23.0 ^b^	43.5 ± 26.6 ^a^	28.1 ± 18.3 ^c^	N.D.
STR (%)	57.3 ± 25.3 ^b^	63.1 ± 26.8 ^a^	49.2 ± 24.2 ^c^	N.D.
ALH (μm)	4.9 ± 1.0 ^b^	2.2 ± 1.0 ^c^	7.6 ± 1.4 ^a^	N.D.
BCF (Hz)	6.2 ± 3.4 ^a^	4.7 ± 3.2 ^c^	5.7 ± 3.6 ^b^	N.D.
n	2815	9111	519	42,306
%	5.14	16.64	0.95	77.27

^a–c^ Different letters represent significant differences (*p* < 0.05) between subpopulations. n: number of sperm; %: percentage of sperm in each subpopulation; N.D: no data; sP: subpopulation. Data are presented as mean values ± SD.

**Table 5 animals-11-03452-t005:** Effect of seminal plasma and its resilience under in vitro capacitation conditions on thawed buck sperm motile subpopulation with respect to age (months), season, and melatonin implant.

Subpopulation (%)	Incubation Media	Autumn		Spring		Spring + Melatonin	
		12	24	18	30	18	30
sP1	T_0_	12.5 ± 1.0 ^1^	14.0 ± 2.7 ^1^	7.9 ± 1.5 ^1^	7.0 ± 1.3 ^1^	7.1 ± 1.2 ^1^	7.7 ± 2.3 ^1^
	Control	6.7 ± 1.7 ^2^	6.4 ± 0.8 ^12^	3.5 ± 0.9 ^2^	3.6 ± 0.9 ^12^	2.9 ± 0.6 ^2^	3.4 ± 0.9 ^12^
	IVF	4.6 ± 1.1 ^ab,2^	6.3 ± 1.6 ^a,12^	1.8 ± 0.2 ^b,2^	1.6 ±0.8 ^b,23^	1.6 ± 0.5 ^b,2^	1.5 ± 0.9 ^b,2^
	IVF + SP	0.0 ± 0.0 ^3^	0.2 ± 0.2 ^2^	0.0 ± 0.0 ^2^	0.0 ± 0.0 ^3^	0.0 ± 0.0 ^3^	1.0 ± 0.6 ^2^
sP2	T_0_	26.3 ± 3.4	34.1 ± 1.9 ^1^	25.2 ± 4.0	25.2 ± 3.4 ^1^	28.0 ± 2.8 ^1^	30.5 ± 2.8 ^1^
	Control	15.2 ± 1.8	14.2 ± 3.0 ^2^	14.6 ± 1.2	14.2 ±1.8 ^12^	10.6 ± 2.0 ^2^	16.7 ± 4.7 ^12^
	IVF	18.7 ± 1.3 ^ab^	25.6 ± 3.2 ^a,1^	15.2 ± 3.4 ^ab^	13.5±1.9 ^ab,2^	11.8 ± 2.1 ^b,2^	8.5 ± 2.0 ^b,2^
	IVF + SP	18.5 ± 8.0	13.2 ± 2.3 ^2^	15.3 ± 3.3	11.0 ± 0.9 ^2^	11.2 ± 3.4 ^2^	7.9 ± 2.9 ^2^
sP3	T_0_	3.2 ± 0.6 ^a,1^	2.4 ± 0.4 ^ab^	0.9 ± 0.4 ^b^	0.9 ± 0.3 ^b,1^	1.1 ± 0.5 ^b^	1.6 ± 0.5 ^ab,1^
	Control	0.5 ± 0.2 ^23^	1.1 ± 0.3	0.7 ± 0.4	0.3 ± 0.2 ^12^	0.3 ± 0.2	0.3 ± 0.2 ^12^
	IVF	0.9 ± 0.3 ^2^	0.8 ± 0.3	0.3 ± 0.2	0.2 ± 0.2 ^12^	0.9 ± 0.4	0.0 ± 0.0 ^2^
	IVF + SP	0.0 ± 0.0 ^b,3^	2.6 ± 1.3 ^a^	0.0 ± 0.0 ^b^	0.0 ± 0.0 ^b,2^	0.2 ± 0.2 ^ab^	0.0 ± 0.0 ^b,2^
sP4 statics	T_0_	58.0 ± 4.1 ^2^	49.5 ± 2.9 ^3^	66.0 ± 5.1 ^2^	66.8 ± 3.9 ^3^	63.8 ± 3.7 ^2^	60.2 ± 4.8 ^2^
	Control	77.6 ± 3.1 ^1^	78.4 ± 3.3 ^12^	81.2 ± 2.1 ^12^	82.0 ± 2.1 ^2^	86.2 ± 2.4 ^1^	79.6 ± 5.4 ^12^
	IVF	75.9 ± 0.8 ^bc,12^	67.3 ± 4.2 ^c,23^	82.7±3.5 ^abc,1^	84.6±1.1 ^ab,2^	85.7 ±2.8 ^ab,1^	90.0 ± 2.9 ^a,1^
	IVF + SP	81.6 ± 8.0 ^1^	84.0 ± 1.7 ^1^	84.7 ± 3.3 ^1^	89.0 ± 0.9 ^1^	88.6 ± 3.5 ^1^	91.1 ± 3.2 ^1^

^a–c^ Different letters represent significant differences (*p* < 0.05) between treatments and age. ^1–3^ Different numbers represent significant differences (*p* < 0.05) between samples within the parameters, T_0_ = time 0, just after sperm washing by single layer centrifugation (SLC); control = modified PBS, IVF: 3 h incubation in in vitro fertilization media; IVF+SP: 3 h incubation in in vitro fertilization media + 20% seminal plasma, sP: subpopulation. Data are shown as mean ± S.E.M.

## Data Availability

The data presented in this study are available on request from the corresponding author.

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
