# Peer review of "The Re-Addition of Seminal Plasma after Thawing Does Not Improve Buck Sperm Quality Parameters"

_animals, 2021, doi:10.3390/ani11123452_

Round 1

Reviewer 1 Report

The current paper depicts the effects of seminal plasma re-addition in post thawed sperm samples from Cabra Blanca de Rasquera goat, which is known to be an endangered breed. The results show that re-addition of seminal plasma does not improve sperm post thaw quality parameters, however, as this breed is an endangered one and not a lot of studies have been conducted on it, this paper can be helpful for future research on this breed and for conservation plans.

The authors have answered all the inquiries from the previous review, and the quality of the paper has significantly improved. However, there are still some questions that need to be addressed.

Lines 76-81. This part should be re-written; there are a lot of syntax mistakes. How about this version:

“However, the seminal plasma of goat semen contains egg yolk coagulating enzymes (EYCE, phospholipase A2) from bulbourethral glands secretions, that interact with egg yolk, and produce toxic substances that affect sperm plasma membrane integrity and motility, and reduce their freezing ability [4,5]”:

Lines 88-100. Please review the English in this part; there is a lot of redundancy.

Lines 103-106. For more homogeneity, please be consistent with your choice of words, use sperm or sperm cells or spermatozoa. In these 3 lines the three words have been used all together. Please choose one word and use it in the entire manuscript.

Line 117. Please correct it to “thawed sperm” instead of “thaw sperm”

Line 164. Why did you choose 20% SP specifically? Did you conduct a preliminary study to determine it or was it used like that in previous studies? Please precise how the optimal concentration was determined.

Lines 210-213. Is it a protocol used in your laboratory or is it a traditional protocol used for goat sperm cryopreservation? Add a reference if possible.

Author Response

Reviewer 1

The current paper depicts the effects of seminal plasma re-addition in post thawed sperm samples from Cabra Blanca de Rasquera goat, which is known to be an endangered breed. The results show that re-addition of seminal plasma does not improve sperm post thaw quality parameters, however, as this breed is an endangered one and not a lot of studies have been conducted on it, this paper can be helpful for future research on this breed and for conservation plans.

The authors have answered all the inquiries from the previous review, and the quality of the paper has significantly improved. However, there are still some questions that need to be addressed.

First of all, the authors would like to thank all the comments and suggestions from the referees which have been very helpful for the authors to improve the present manuscript

Lines 76-81. This part should be re-written; there are a lot of syntax mistakes. How about this version:

“However, the seminal plasma of goat semen contains egg yolk coagulating enzymes (EYCE, phospholipase A2) from bulbourethral glands secretions, that interact with egg yolk, and produce toxic substances that affect sperm plasma membrane integrity and motility, and reduce their freezing ability [4,5]”:

Lines 88-100. Please review the English in this part; there is a lot of redundancy.

Lines 103-106. For more homogeneity, please be consistent with your choice of words, use sperm or sperm cells or spermatozoa. In these 3 lines the three words have been used all together. Please choose one word and use it in the entire manuscript.

Line 117. Please correct it to “thawed sperm” instead of “thaw sperm”

All the above inquiries have been corrected in the last version of the manuscript

Line 164. Why did you choose 20% SP specifically? Did you conduct a preliminary study to determine it or was it used like that in previous studies? Please precise how the optimal concentration was determined.

The authors did not conduct a preliminary study to determine the percentage of seminal plasma added to the incubation media. The decision of the authors to adopt the inclusion of adding seminal plasma at 20% was due a previous works done by Rovegno et al., (2013) on ram using same concentration which showed better results when SP was added at a concentration of 20%.

[19]: Rovegno, M.; Feitosa, W. B.; Rocha, A. M.; Mendes, C. M.; Visintin, J. A.; D’Avila Assumpção, M. E. O. Cell Tissue Bank. 2013, 14 (2), 333–339.

Lines 210-213. Is it a protocol used in your laboratory or is it a traditional protocol used for goat sperm cryopreservation? Add a reference if possible.

The protocol used for goat sperm cryopreservation could be considered as a traditional protocol except the non-penetrant cryoprotectant used. The authors have been working in previous studies with the use of powdered egg yolk, instead of the traditional fresh egg yolk used in most of the laboratories. Reason due to the low biological risk, increase homogeneity and less cumbersome in the preparation of the extenders. (For more information see reference [21]: Tabarez et al, 2017. Small Ruminant Research 149 (2017) 91–98).

Reviewer 2 Report

After reading the author’s response to my comments, I could understand the scientific soundness of the research, so I suggest they include in the article the same explanation they gave me: “In fact, the final objective was to find a new method during the thawing process which can be applied in all semen bank doses regardless of the freezing protocol used in order to increase fertility rates after AI is performed. Traditionally, since egg yolk has been used to create semen bank from different goat breeds, our aim was to try to propose a method to improve this thawed sperm quality regardless of the origin or freezing protocol of buck semen.”

Author Response

Reviewer 2

After reading the author’s response to my comments, I could understand the scientific soundness of the research, so I suggest they include in the article the same explanation they gave me: “In fact, the final objective was to find a new method during the thawing process which can be applied in all semen bank doses regardless of the freezing protocol used in order to increase fertility rates after AI is performed. Traditionally, since egg yolk has been used to create semen bank from different goat breeds, our aim was to try to propose a method to improve this thawed sperm quality regardless of the origin or freezing protocol of buck semen.”

The authors agree the suggestion of the reviewer and have modified the manuscript hoping to make it more understandable and stronger from a scientific point of view. The authors want to thank to the reviewer, his/her help in improving the explanation of the main focus of the present study.

Reviewer 3 Report

The authors addressed all concerns raised in my initial review, though I don't necessarily agree with all their reasoning. But these may be more my personal concerns and should not keep the manuscript form being published. The inclusion of the follow-up experiment mentioned by authors would have been a useful addition to this manuscipt.

Author Response

Reviewer 3

The authors addressed all concerns raised in my initial review, though I don't necessarily agree with all their reasoning. But these may be more my personal concerns and should not keep the manuscript form being published. The inclusion of the follow-up experiment mentioned by authors would have been a useful addition to this manuscript.

The authors agreed with the reviewer comments about the inclusion of the follow-up experiment, but at the same time they were also afraid of adding more factors on the present study making the whole manuscript very confusing and excessively long as it was a broad research project.

This manuscript is a resubmission of an earlier submission. The following is a list of the peer review reports and author responses from that submission.

Round 1

Reviewer 1 Report

The research article proposed here evaluates the effects of season, age, seminal plasma, and capacitation media on buck sperm kinematic parameters, acrosome and DNA integrity, and vitality. A lot of results are discussed in this paper and the experiments were well conducted. However, there are some serious flaws in this manuscript that needs to be addressed, as there is a lack of clarity when it comes to the purpose of the study, which makes the reading and understanding of the manuscript harder. I suggest the authors to revise the title of the manuscript, aim of the study, and rearrange the rest of the manuscript accordingly.

Specific comments

Simple summary and Abstract: it lacks clarity and needs heavy English editing. Furthermore, the aim of the study is not clearly stated and is different from the one in the Abstract and the one in the Discussion:

  1. Are you working on a new Thawing protocol or are you working on the enhancement of Post-thawed sperm quality parameters? These two things are totally different, and you need to explain exactly what is the aim of your study.
  2. There seems to be another experimental design involving age, breeding and non-breeding periods sperm quality, which was mentioned neither in the simple summary nor in the title. Please be consistent with the title and the aim of your study.
  3. In the discussion, the aim of the study seems to be the assessment of functional and kinetic characteristics of frozen-thawed buck sperm during an incubation period to mimic the environment of the female reproductive tract.

Introduction: This part as well needs heavy English editing. In addition, the contents of the introduction and the aim of the study do not match. I will state the reasons here:

  1. There is little to no information about the role of SP and why its addition to post-thawed sperm under in vitro capacitation conditions is necessary.
  2. The title states: “Seminal plasma effect on post-thawed buck sperm from Cabra Blanca de Rasquera breed and its resilience under in vitro capacitation conditions”, which means the introduction should cover: the current state of sperm cryopreservation in buck, a definition of capacitation and its importance, seminal plasma and its components, effects, and importance, and lastly your hypothesis on how seminal plasma can act on sperm under in vitro conditions.
  3. There is a paragraph about the effect of age, and season on sperm quality, while the aim of the study is supposed to be the evaluation of SP effects on thawed sperm.

How does freezing and thawing affect sperm ultrastructure and function, and by which mechanism seminal plasma might be able to counteract these deleterious effects?

Lines 60-74. Please elaborate more in this part. You need to explain what are cryo-injuries and when do they happen? What does the EYCE do to sperm and by which mechanism do they alter sperm viability and motility. Scientists who are not familiar with this field cannot understand the impact of cryo-injuries and EYCE, hence the importance of the introduction part.

Lines 99-105. There is a lot of redundance in these two paragraphs when they can be resumed into one small sentence. It is a known fact among veterinarians and theriogenologists that goats` breeding season is autumn, and that sperm quality and concentration fluctuate depending on the season. You can just state that male goats breeding season is in autumn and that semen quality is higher in that season. Just make it simple.

Materials and methods: The experimental designs that were chosen for this study are very confusing. Please provide an additional segment explaining the experimental designs and their aims.

Discussion:

Lines 474-478. “we may suggest that the commercial IVF media used was not efficient enough to induce in vitro capacitation and subsequently acrosome reaction or, alternatively, the incubation time was insufficient to see more capacitation-like changes”. This is a clear contradiction to the explanation given in the previous paragraph in lines 459-462: “In other words, the IVF media may have induced an increase in live acrosome reacted sperm which eventually dies if fertilization does not take place [26]”.

General comments:

Line 21. Conservation programmes for semen cryopreservation and the development of a genetic sperm bank…. Seems more correct.

Line 28. To provide not to provides. Same for the other verbs.

Line 76. Being a good source of nutrient is not a role in key processes. Please remove this example

Line 79. Please change the font size.

Lines 207-210. Why did you choose these centrifugation conditions? Please add a reference.

Line 214. “samples were diluted in the various incubating media” what do you mean by this phrase?

Line 231. Please correct this part with: Using a final concentration of 1 nM of SYBR-14

Line 232. Please correct to: 2 analysis were performed.

Line 267. Please correct it to: DNA integrity assessment was performed immediately after etc,….

Lines 302-303. Please change into : Total motility (TM, %), progressive motility (PM, %).

Line 312. Please correct to: progressive (without capital letter).

Line 330. Please correct to: An ANOVA test was applied.

Line 331. Please correct to: GLM (General Linear Model)

Line 332. Please correct to: with a Tukey test

Line 345. What do you mean by “Not always significantly”. Please refrain from using adverbs of frequency when explaining/showing scientifical results. You have to show which results were significant and which ones were not, and comment on the tendencies seen in the non-significant results.

Table 1. Please edit the table to show the age factor in the table more clearly. It is not shown INSIDE the table that those random numbers in the second line are the ages of the males.

Line 401. Please correct to: were significantly reduced

Line 418. Please separate the two sentences when the subject is different: motile subpopulations. We observed that… etc.

Line 449. Please correct to either: during a post-thaw incubation period, or: The functional kinetic characteristics of frozen-thawed buck sperm during an incubation period.

Line 486. “Results are” please change the font size here.

Reviewer 2 Report

Abstract

Authors should include results of mitochondria activity.

Introduction

The introduction needs more background. If there is a negative interaction between egg yolk and seminal plasma enzymes, and seminal plasma is important for different reasons, why authors did not choose another diluent, without egg yolk, instead of removing seminal plasma? If there are articles comparing diluents for cryopreservation with and without egg yolk, and it was proven that the ones containing egg yolk are the best for cryopreserving buck sperm, the authors should include these references in the introduction. They need to explain their choice, since it is much easier to try another diluent for cryopreservation than to remove seminal plasma before freezing, just to add it again after thawing. Also, it is more scientifically sound to try to improve cryopreservation process than to try reverting cold shock. In this regard, there is only one citation about reverting cold shock (article 31, from 2000), presented in the discussion (page 16, line 488). The authors should look for more articles and include this background in the introduction.

Results

The authors did not present data from semen analyses before freezing (fresh semen). It is important to know semen quality before freezing, in order to know how much it changed after cryopreservation, and also to know how much it could have improved after the post-thaw treatments.    

I suggest using “plasma membrane integrity” instead of “viability” (page 8, line 342) throughout the article (text and tables).

Discussion

Authors attribute reduction in motility to mitochondria ageing (page 17, line 533), however their results show very low mitochondria inactivity in all treatments (page 8, line 352).

Reviewer 3 Report

The experiment described in the manuscript aimed to identify benefits (expressed in in vitro semen quality parameters) derived from adding seminal plasma to frozen-thawed buck semen. However, in my view the authors compromise the evaluation of seminal plasma by adding it to a commercial fertilization medium designed to influence capacitation.  Any observations made are hence confounded by an interaction with the IVF medium. At a minimum seminal plasma should also have been added as a PBS+seminal plasma treatment, or ideally included a group with seminal plasma as the sole incubation medium. Any discussion of seminal plasma effects (lines 34-36; lines 49-55; lines 514-517; lines 582-585; lines 589-595) have therefore be viewed with great caution.

A second concern is the use of buck age as a variable in this experiment. Firstly, the age of buck is confounded by time of collection.  Secondly, a wider range in buck age would have been more meaningful (young vs. mature), while 1 and 2 year old bucks would both be considered young animals. Finally, if such narrow ranges are considered for age the comparison Autumn and Spring collections in this study would be confounded by age.

A final issue of concern is the presentation of results. The format of the tables does not really allow to assess the main effects.  The tables would be more useful if age, season and melatonin treatment were contrasted directly.

The following minor points should also be considered:

Line 148: addition of powdered egg yolk would be ‘wt/v’?

Line 153: more detail is needed describing ‘intensive buck management’.

Lines 157/159: it is not clear what ‘(n=6)’ refers to here.  If it refers to the number of collections, and these are considered the replications, that that should be spelled out.

Lines 172:  was there any pre-freezing quality assessment of the extended semen?

Line 176: replication in this case refers to day of collection?  See comment earlier.

Line 205: what was the rationale/data used for seminal plasma inclusion at 20%.